# Environmental Chamber Characterization of an Ice Detection Sensor for Aviation Using Graphene and PEDOT:PSS

**DOI:** 10.3390/mi15040504

**Published:** 2024-04-07

**Authors:** Dario Farina, Marco Mazio, Hatim Machrafi, Patrick Queeckers, Carlo Saverio Iorio

**Affiliations:** 1Centre for Research and Engineering in Space Technologies (CREST), Department of Aero-Thermo-Mechanics, Université libre de Bruxelles, 1050 Brussels, Belgium; dario.farina@ulb.be (D.F.); patrick.queeckers@ulb.be (P.Q.); carlo.iorio@ulb.be (C.S.I.); 2Department of Industrial Engineering, University Federico II of Naples, 80138 Napoli, Italy; marcomazio98@gmail.com; 3UFR Physique, Sorbonne Université, 75005 Paris, France

**Keywords:** aerospace icing prevention, risk mitigation in aviation, graphene-based sensors, dynamic ice sensing, de-icing and anti-icing technologies, conductive polymers in sensing, smart sensing technologies, PEDOT:PSS polymers, 2D materials in sensing applications, environmental sensing in aerospace, micromachines, phase transition detection

## Abstract

In the context of improving aircraft safety, this work focuses on creating and testing a graphene-based ice detection system in an environmental chamber. This research is driven by the need for more accurate and efficient ice detection methods, which are crucial in mitigating in-flight icing hazards. The methodology employed involves testing flat graphene-based sensors in a controlled environment, simulating a variety of climatic conditions that could be experienced in an aircraft during its entire flight. The environmental chamber enabled precise manipulation of temperature and humidity levels, thereby providing a realistic and comprehensive test bed for sensor performance evaluation. The results were significant, revealing the graphene sensors’ heightened sensitivity and rapid response to the subtle changes in environmental conditions, especially the critical phase transition from water to ice. This sensitivity is the key to detecting ice formation at its onset, a critical requirement for aviation safety. The study concludes that graphene-based sensors tested under varied and controlled atmospheric conditions exhibit a remarkable potential to enhance ice detection systems for aircraft. Their lightweight, efficient, and highly responsive nature makes them a superior alternative to traditional ice detection technologies, paving the way for more advanced and reliable aircraft safety solutions.

## 1. Introduction

The challenge of detecting ice during a flight in aviation has been a focal point of research due to its critical implications for flight safety, efficiency, and performance [1]. This phenomenon, which poses a considerable challenge in various flight conditions, primarily occurs when supercooled water droplets in the atmosphere come into contact with the aircraft’s surfaces, leading to ice formation. Additionally, ice accumulation on aircraft can result from a variety of atmospheric conditions, including the freezing of rain upon contact and the transformation of water vapor into ice under cold, dry conditions. These processes are most commonly found in clouds or precipitation zones with temperatures below 0 °C but can also occur in clear air conditions characterized by high moisture levels and conducive temperatures for icing. The formation of ice on an aircraft significantly disrupts airflow over wing surfaces and control elements, altering aerodynamic properties, increasing drag, and decreasing lift. This not only compromises the aircraft’s ability to maintain altitude and velocity but also adds substantial weight, impacting performance and efficiency. Furthermore, the effect on control surfaces and propulsion systems critically limit a pilot’s maneuverability and control, especially during takeoff and landing phases [2,3]. In response, the development of ice detection systems has become a key area in aviation technology. Traditional systems, ranging from mechanical methods to thermal and optical methods, have provided foundational information but often face limitations in terms of response time, sensitivity, and applicability under diverse flight conditions [4,5].

Recent advancements in material science have ushered in a new era of potential solutions, with graphene-based technologies at the forefront [6,7]. Graphene’s exceptional conductivity, flexibility, and strength make it an ideal candidate for sensitive and accurate ice detection. Studies have shown promising results in utilizing graphene’s properties for real-time ice formation detection [8].

One promising advancement in ice detection systems is the development of an aircraft sensor fault detection system based on the Steady-State Least Mean Squares (LMS) algorithm, which enhances the reliability of sensor networks in aviation by efficiently detecting and mitigating Byzantine attacks within wireless sensor networks [9]. Additionally, the exploration of aviation sensor performance evaluation methodologies has provided valuable insights into improving sensor reliability and effectiveness in challenging aviation environments, paving the way for more accurate and reliable ice detection technologies [10].

Micro-electromechanical system (MEMS) technology has been widely explored for ice detection in aviation. Varadan et al. [11] provide an overview of MEMS-based sensors and their potential applications in smart structures and systems. The authors discuss the advantages of MEMS technology, including the small size, low power consumption, and high sensitivity, which makes such systems ideal for ice detection in aviation. However, the paper also highlights the challenges and limitations of MEMS-based sensors, such as their susceptibility to environmental factors and the need for proper calibration and testing.

Another interesting study is the work by Strijhak et al. [12], which presents a neural-network-based approach for predicting ice shapes on airfoils using computational fluid dynamics simulations. The researchers trained a neural network using data from iceFoam simulations to predict the ice shape on an airfoil under various icing conditions. The results of the study show that the neural network approach is able to accurately predict the ice shape with a high degree of accuracy, demonstrating the potential of this approach for enhancing ice detection and mitigation in aviation.

Further enhancing ice detection, the microstrip patch antenna method represents a leap in de-icing technology. It detects ice through frequency shifts caused by near-field disruptions, offering thermal stability across a broad temperature range, which underscores its potential in advancing ice detection efficiency [13].

Despite these advances, the field continues to face challenges. There exists a spectrum of hypotheses on the most effective and practical approach to ice detection. The divergence lies mainly in balancing the sensitivity, reliability, and cost-effectiveness of the systems in varying environmental conditions [14,15].

This research aims to critically evaluate the performance of graphene-based sensors within an environmental chamber designed to simulate a range of atmospheric conditions pertinent to ice formation. This approach offers a novel perspective on understanding the capabilities and limitations of graphene sensors compared to existing technologies. The results are expected to contribute significantly to the current understanding of ice detection technologies, providing valuable insight for future advancements.

In summary, this research not only contributes to the ongoing discourse on the advanced detection of in-flight ice but also posits the potential of graphene-based technology as a transformative solution in enhancing aircraft safety. Our graphene-based sensor utilizing PEDOT:PSS is primarily aimed at aviation safety, offering real-time detection of ice on aircraft surfaces. Beyond aviation, it has potential applications in telecommunications and road transport. Its accuracy, however, may be influenced by extreme environmental conditions, with constraints on its performance near 0 °C. This innovation promises significant advancements in ice detection technologies across various industries.

## 2. Experimental Setup and Methodology

In this section, we describe the development of the sensors used for the experiments in the ice detection system. Each sensor comprises a sensing part responsible for signal generation and a conductive part for signal transport to the data logger.

### 2.1. PEDOT:PSS: A Conductive Polymer for Sensing

This study explores the use of PEDOT in thermoelectric devices used for ice detection, emphasizing its flexibility, eco-friendliness, and cost-effectiveness. Despite its atmospheric sensitivity and moderate efficiency, doping with PEDOT:PSS improves conductivity through redox reactions, affecting the structure. PEDOT’s excellent conductivity and stability, along with PEDOT:PSS’s role as a mixed conductor, are crucial, although GOPS (3-glycidyloxypropyl)trimethoxysilane, which is often used to increase the stability of PEDOT:PSS through crosslinking, may, however, lower the conductivity due to its effects on the material’s structural properties [16]. Its structure facilitates charge mobility, with humidity affecting charge transport modes. These findings are vital for stating the possibility of using it in flight conditions [17,18]. The principle of ice detection by the sensor involves using PEDOT:PSS, a polymeric mixed ionic–electronic conductor that displays a significant increase in electrical resistance during the phase transition from liquid water to solid ice. This change is attributed to the morphology and electronic transport in PEDOT being affected by the freezing event, as the absorbed water in the PSS-rich phase expands upon forming ice crystals.

### 2.2. Graphene-Based Sensors

Graphene is a single layer of carbon atoms arranged in a two-dimensional honeycomb lattice, and it is known for its remarkable strength, flexibility, electrical, and thermal conductivity. Graphene’s integration into 3D structures enhances sensing systems, offering low thermo-mechanical stress and easy incorporation into complex shapes [19]. Its exceptional conductivity increases operational efficiency in applications such as ice detection in aircraft, significantly reducing energy consumption. The compatibility of graphene with various materials, including polymers and fibers, broadens its application scope. The consistently low resistance of the material simplifies PEDOT:PSS resistance analysis, improving system evaluations. A particular graphene electrode configuration was chosen for its effectiveness in identifying ice, highlighting graphene’s utility in innovative sensor configurations. This design, which is detailed in figures, showcases the integration and application of graphene electrodes within the detection system, emphasizing their dimensional precision and fabrication specifics, which are crucial for the sensor’s performance. Our graphene-based sensor utilizes graphene’s unique electrical properties, including sensitivity to changes in electrical conductivity and capacitance, to detect ice formation. When ice forms on the sensor’s surface, it disrupts the charge distribution and electrical pathways of the graphene, a change that is rapidly detected and quantified for precise icing event identification.

### 2.3. Sample Creation

For the sensing element, PEDOT:PSS is employed due to its hygroscopic properties, conductivity, water solubility, and elasticity [6].

The fabrication process for PEDOT:PSS (ratio of 1:2.5) sensing films involved the careful formulation of the polymer mixture, starting with Clevios PH1000 from Heraeus Holding GmbH (Leverkusen, Germany), which was filtered through a 0.45 µm polyvinylidene fluoride (PVDF) filter for purity. GOPS was then integrated for mechanical stability, ensuring that the films maintained their integrity upon exposure to moisture. This step was crucial for enhancing the longevity and reliability of the sensors under various environmental conditions. The solution was subjected to thorough mixing and sonication for 5–10 min to guarantee a homogeneous mixture, setting the stage for optimal film deposition. This method underscores the importance of precision in creating sensors that exhibit both high electronic and ionic conductivity, which is crucial for their functionality in detecting environmental changes such as ice formation. The interaction between the inherent conductive properties of the polymer and external factors such as humidity is a key consideration in the design and application of these sensing films, with the aim of leveraging their unique charge transport mechanisms for efficient and accurate sensing capabilities [6].

The capacity to absorb water is crucial for the detector’s operation. The sensing element provides a signal by detecting changes in resistance as the absorbed water transitions from the liquid to the solid ice state. Graphene was produced as GS50-type graphene ribbons by NANESA SRL (Arezzo, Italy), with dimensions of 40 mm × 5 mm × 0.10 mm. In this research, GS50 denotes the graphene paper utilized as a crucial element of our sensor configuration [20].

Silicone rubber, also known as PDMS, which was trimmed to specific sizes, provided a pliable foundation that was crucial for integrating the system into an airfoil. Prepared graphene strips that were cut to exact dimensions formed the conductive routes. Copper adhesive tape secured these strips, ensuring robust electrical linkages. A carefully placed droplet of PEDOT:PSS, applied through drop-casting, connected the strips, transforming them into a unified conductive route once it is dried.

Following this, the samples underwent a 24 h drying period at room temperature to guarantee the durability of the conductive link. The completed version of the sensor is illustrated in Figure 1. Resistance tests confirmed the sensor’s reliability prior to its use.

The concluding phase entailed attaching wires to the conductive copper tape, which acted as a bridge between the graphene and cables, and linking this setup to a data acquisition unit, the Agilent 34970A (Agilent Technologies, Santa Clara, CA, USA). This connection allowed for the continuous observation and analysis of resistance, linking resistance changes to phase transitions caused by airflow. The finished sensors, as shown in Figure 1, were distinguished by their adaptability, compact size, and light construction. These attributes allow for their unobtrusive attachment to aircraft exteriors, permitting a collaborative sensor network for pinpoint ice detection. This method is in line with the broader objective of enhancing energy efficiency through focused de-icing bolstered by accurate detection techniques. A critical aspect to highlight is the sensor’s adaptability, offering the flexibility to modify its size according to the intended use area while maintaining its capabilities for detecting water and ice.

### 2.4. Environmental Chamber Setup

The experiments were carried out within a controlled environmental chamber built in our facilities (Université Libre de Bruxelles, Bruxelles, Belgium). The environmental chamber allowed for the manipulation of the inside temperature, the temperature of the injected droplets, and the humidity levels, closely replicating a range of climatic scenarios relevant to ice formation on an aircraft. This chamber served as a controlled test bed crucial for evaluating the performance of the graphene-based ice detection sensors.

The schematic diagrams depict the environmental chamber designed for the simulation, control, and monitoring of environmental conditions. Figure 2a centers on the chamber itself, outfitted with temperature sensors of type T and a top-view camera (JAI 500B) for visual monitoring, as well as a side camera (JAI 5000) for a side camera view. Alongside this, a cooling mechanism with distinct inlets and outlets provided the capability for detailed temperature adjustments. A second thermal bath supplied the capacities of temperature adjustment (between +1 ${\;\;}^{\circ }$C and +40 ${\;\;}^{\circ }$C) of the droplets injected.

The climate chamber consisted of a cubic metal body cooled by a refrigerant fluid circulating within a serpentine of pipes arranged around the chamber walls. To enhance the thermal insulation, the entire chamber was covered with 40 mm thick polyester that was 800 mm high.

A thermal bath was connected to the serpentine, enabling precise temperature control within the chamber. The chamber could maintain a minimum temperature of −20 ${\;\;}^{\circ }$C, allowing for the simulation of a cryogenic environment.

Water was injected into the chamber through a syringe pump controlled by a LabView unit. This setup facilitated the injection of a defined volume of water in the form of small droplets. Typically, 24 µL droplets were released during the experiments to study the effects of a specific quantity of water on PEDOT:PSS, which translated to an estimated diameter of approximately 5 mm. The pump mechanism was approximately 30 cm above the sample plate.

In the setup, T-type copper–constantan thermocouples were used.

Each sample was mounted on a Tec1-12715 Peltier element measuring 40 × 40 × 3.3 mm. This element operated on the Peltier effect, generating a temperature difference between its two sides when a voltage was applied. The cold side could reach temperatures below room temperature, allowing ice formation, and T${\;\;}_{cold}$ was the temperature measured on the cold side of the Peltier. A heat sink was attached to the hot side to dissipate heat rapidly.

A 60-watt heater was employed to facilitate the melting of ice formed on the sensor after each experiment.

Adjacent to the environmental chamber were an acquisition system and a circuit board, serving as the system’s computational and control hub. Figure 2b expands on this by showing an Arduino board connected to various components, including an LCD displaying temperature and humidity values and providing a user interface for real-time data monitoring. The humidity level within the facility was controlled by activating a humidifier (to increase the humidity) or turning on two small fans to circulate air and reduce the humidity.

For data acquisition, the ice detector was connected to an Agilent 34970A (Agilent Technologies, Santa Clara, CA, USA) data logger, which read three temperature values (two from thermocouples on chamber walls and one from the sensing element of the detector) and the internal resistance of the sample, including the PEDOT, graphene strips, and electrical connections. This instrument combined precision measurement capabilities with flexible signal connections and was capable of logging and data acquisition. During the tests in the climate chamber, data were acquired at a frequency of 5 Hz.

Additional elements such as a humidifier and a heater coupled with a fan underscored the system’s ability to manipulate both temperature and humidity levels within the chamber. Fans positioned around the chamber facilitated air circulation, which was essential for maintaining consistent conditions throughout the space.

### 2.5. Experimental Procedure

The objective of the experiments was to create icing environments to study the signals generated by the ice detection system and to analyze how these signals changed under varying external conditions, such as the chamber temperature. Such a study is crucial for characterizing this detection technology, which will find its primary application in the subsequent wind tunnel tests that will be conducted [8].

The experiment began by setting the temperature parameters and the number of iterations. The acquisition system was started for video and data recording, followed by data acquisition. Water droplets were injected, and cycles of heating and cooling occurred.

When a droplet from the syringe was injected from above (Figure 3), it fell onto the sensing element of the sample and was absorbed by the PEDOT. When the cold side reached a temperature below 0 ${\;\;}^{\circ }$C, the water froze, and ice formed on the sample. After this process was measured and recorded, the ice was melted by increasing the temperature of the Peltier element or a fan in preparation for the next cycle.

The maximum temperature range achievable with the Peltier cell used was 60 ${\;\;}^{\circ }$C. The temperature was closely monitored using the thermocouples described above. The humidity level inside the chamber was maintained at a constant 90%.

Table 1 provides a matrix of the climatic chamber tests, indicating the range of minimum T${\;\;}_{cold}$ temperatures for each experiment along with the number of iterations. Experiments at higher temperatures were observed to require more time to form ice.

Furthermore, the impact of the chamber temperatures was evaluated, although most of the experiments were conducted at positive T${\;\;}_{chamber}$ temperatures.

In Figure 4, a representative series of cycles that were executed is presented to illustrate the subtle influence of temperature variation on the resistance, which paled compared to significant changes, of the order of kilo or mega ohms (KΩ), observed during ice formation, as will be discussed later. It is to be noted that a temperature sensor was strategically positioned directly above the PEDOT surface to monitor temperature fluctuations upon droplet impact precisely. The fluctuations observed in T${\;\;}_{pedot}$ were attributed to heat exchange phenomena, while abrupt resistance fluctuations were mainly attributed to the reduced conductivity of ice compared to water, a characteristic that became evident during the phase transition [6,7]. In addition, the observed trend of increasing resistance with each cycle could be further explained by the gradual decrease in water volume on the sensor’s surface, which was likely due to evaporation. This resulted in the resistance stabilizing at a higher value for each subsequent cycle, reflecting the lower conductivity as a result of the reduced water content. This complemented the initial observation that while temperature variation subtly influenced resistance, the more substantial changes were primarily attributed to the conductivity differences between ice and water during phase transitions.

## 3. Results

The characterization of the ice detection system was performed by linking the changes in internal resistances to the phase changes in the water absorbed by the sensing element of the detector.

Each sample had a similar internal resistance (order of magnitude of $10^{4}$ kΩ), eliminating the need to use a normalized resistance.

The volume of each drop was estimated at 24 µL, and the injected water was at ambient temperature. The cycles performed could be observed from the temperature trend provided by the thermocouple in contact with the sensing element.

The amplitude of the signals and their dependence on the chamber conditions were investigated. The figures show the resistance and temperature patterns during droplet injection, ice formation, and ice melting.

Figure 5 displays the dynamic process of ice formation over three cycles, characterized by an initial drop in resistance and a rise in T${\;\;}_{pedot}$ values. This initial change is attributed to the impact of a water droplet. As time passed, the resistance increased sharply, as shown in the zoom of Figure 6, indicating the phase transition from liquid to solid as the droplet froze. The cyclical pattern of these changes was consistent across the events, demonstrating the sensor’s ability to detect and respond to successive icing conditions with high repeatability. Figure 7 and Figure 8 show the same with different values of T${\;\;}_{cold}$. It appeared that the increase in resistance was lower as T${\;\;}_{cold}$ increased. The values of these resistance increases are shown in Table 2.

Furthermore, Figure 9 shows the resistance behavior of an ice detection sensor during the phase change within the environmental chamber, demonstrating the effect of the ambient temperature on the amplitude of the ice signal. As the chamber’s temperature was systematically lowered, the amplitude of the resistance signal increased, which was indicative of the enhanced sensitivity of the sensor to the phase change at colder ambient temperatures. This implies that the colder the chamber when keeping the same temperature of the droplet and plate for the cycles, the more distinct the resistance change, which is critical for precise ice detection. The comparison between the two graphs at different chamber temperatures (15 °C and 10 °C) highlights the sensor’s response to environmental changes, reinforcing the importance of temperature control in the calibration and operation of ice detection systems. This analysis confirms the sensor’s reliability and its potential application in environments where temperature variations play a crucial role in the formation of ice.

Another relevant remark concerns the signals obtained upon the injection of the water. A correlation can be observed between the amplitude of the signals and the minimum Peltier temperature, as shown in Figure 10. In particular, as reported in Table 3, the amplitude of the resistance drop was greater when $T_{\mathrm{cold}}$ was lower. In Table 3, Δ*R*${\;\;}_{water}$ is defined as the difference between the final and the initial values of the resistances, respectively, after and before injection. Therefore, the modulus was considered, since ΔR${\;\;}_{water}$ is always negative by definition. The reason is that the resistance *R* and the temperature *T* are inversely proportional quantities. As a consequence, when the temperature is lower, the value of the resistance before the injection is higher and, therefore, the amplitude is bigger. Either way, a minimum value exists below which the resistance cannot drop. This is shown in Figure 10 to be around 40 kΩ.

It can also be observed that after the injection signal, the resistance tended to increase with a rate that was higher at lower temperatures because of the intrinsic behavior of the PEDOT.

## 4. Conclusions and Discussion

In this study, a graphene-based sensor was developed to enhance the detection of ice formation. Utilizing an environmental chamber, we systematically varied environmental conditions, such as the temperature and humidity. The sensor’s response to these changes was meticulously recorded, and the resistance behavior was analyzed, indicating phase transitions from the liquid to the solid states. This methodical approach allowed us to explore the potential of the sensor for real-time ice detection in various applications. The observed increase in ice signals’ amplitudes with decreasing minimum temperature and the associated effects of the chamber temperature underscore the sensor’s sensitivity to environmental conditions. This sensitivity is critical in applications where accurate ice detection is paramount, such as in aviation and climate research. The study suggests a strong correlation between the amplitude of ice signals in our sensor and the phase-change properties of water under varying thermal conditions. As the chamber temperature decreases, the resistance before droplet injection is higher, leading to more pronounced changes upon freezing.

The signal behavior can be summarized as follows:When a droplet touches the sensing element (PEDOT:PSS) of the detector, there is a drop in the internal resistance of the test samples.A steep increase in resistance occurs when the absorbed water freezes, obstructing the passage of electric charges through the graphene strips due to the formation of ice crystals inside the PEDOT.An abrupt drop in resistance occurs when the ice melts at a positive local temperature, resulting in a signal similar to the one acquired at the injection. Both signals are classified as “water” signals.

Our research focuses on testing graphene-based sensors in an environmental chamber that simulates a variety of climatic conditions that an aircraft could experience during flight. The results indicate that the developed sensors exhibit heightened sensitivity and rapid response to subtle changes in environmental conditions, especially the critical phase transition from water to ice. This sensitivity is crucial for detecting ice formation at its onset, which is essential for aviation safety. In terms of durability, a few studies have evaluated the long-term performance of graphene-based sensors. One study found that graphene sensors showed no loss of sensitivity after being exposed to harsh conditions, indicating their potential for long-term use in aviation. However, more studies are needed to evaluate the durability of these sensors in comparison to other alternatives. In terms of cost-effectiveness, the cost of graphene-based sensors is relatively higher than that of traditional sensors. However, the lightweight, efficient, and highly responsive nature of the graphene sensors makes them a superior alternative to traditional ice detection technologies, which could ultimately reduce costs associated with aircraft maintenance and safety. This research significantly contributes to the field of micromachines by pioneering the application of graphene- and PEDOT:PSS-based sensors for ice detection in aviation. It introduces a novel approach that enhances the sensitivity, reliability, and efficiency of ice detection technologies. By leveraging the unique properties of graphene and PEDOT:PSS, the study not only addresses a critical safety concern in aviation but also opens avenues for the development of lightweight, high-performance micromachines with broad implications across various industries. In conclusion, this study reveals graphene-based sensors’ broad applicability and transformative potential in detecting ice formation across multiple sectors. Beyond enhancing aviation safety, these sensors can significantly impact wind turbine efficiency by preventing ice accumulation, improving road safety by early ice detection, and supporting maritime navigation and agriculture by mitigating ice and frost risks. This wide range of applications underscores the innovative and versatile solutions that graphene-based sensors offer to industries confronting challenges related to ice formation.

Future investigations could explore the optimization of sensor materials to extend the lower threshold of detection, enhancing the sensitivity and range of operation. Additionally, the integration of these sensors with real-time data analysis and machine learning could predict icing events, allowing preemptive actions to be taken in industrial and aeronautical applications. Such advances could lead to more efficient de-icing systems, reduced material wear from ice accumulation, and improved overall safety.

## 5. Patents

International patent n${\;\;}^{\circ }$ https://www.ulb.be/en/technology-sharing/device-and-method-for-detecting-ice-formation-technology-offer WO2022162068A1 (accessed on 6 April 2024)/https://worldwide.espacenet.com/patent/search/family/074346829/publication/EP4036565A1?q=pn%3DEP4036565A1 EP4036565A1(accessed on 6 April 2024).

## Figures and Tables

**Figure 1 micromachines-15-00504-f001:**
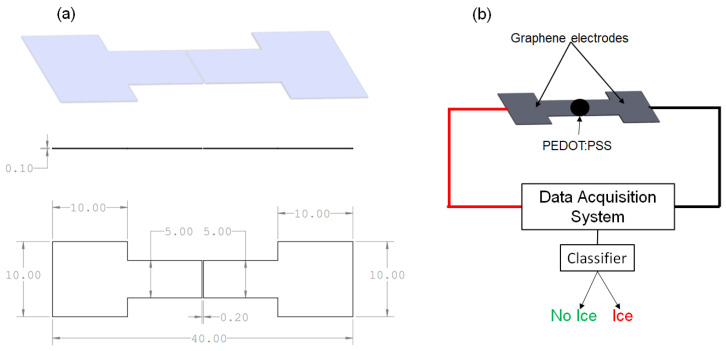
(**a**) Sketch of the electrodes engineered and utilized in all characterization studies. Measurements are depicted in millimeters. (**b**) The unified system featuring the graphene electrodes, PEDOT:PSS detection layer, data gathering system, and interpretable output display.

**Figure 2 micromachines-15-00504-f002:**
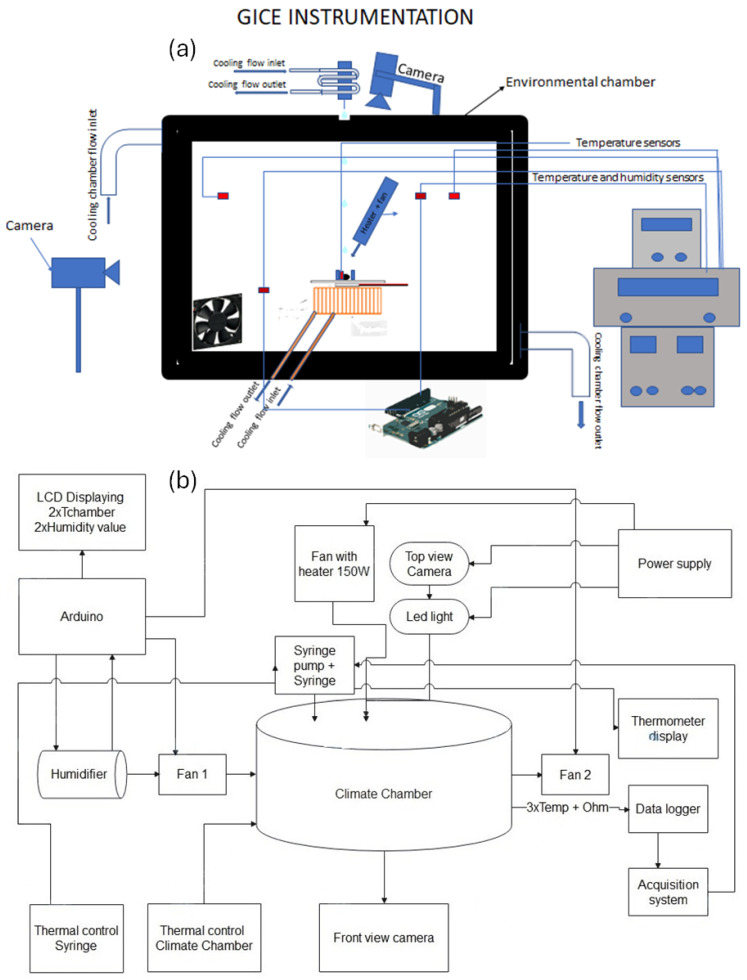
(**a**) Diagram of the setup. The environmental chamber is depicted with the main components. (**b**) A schematic of the climate chamber setup used for the ice detection experiments, featuring an Arduino-based control system, humidity and temperature regulation, and real-time monitoring capabilities via cameras and sensors.

**Figure 3 micromachines-15-00504-f003:**
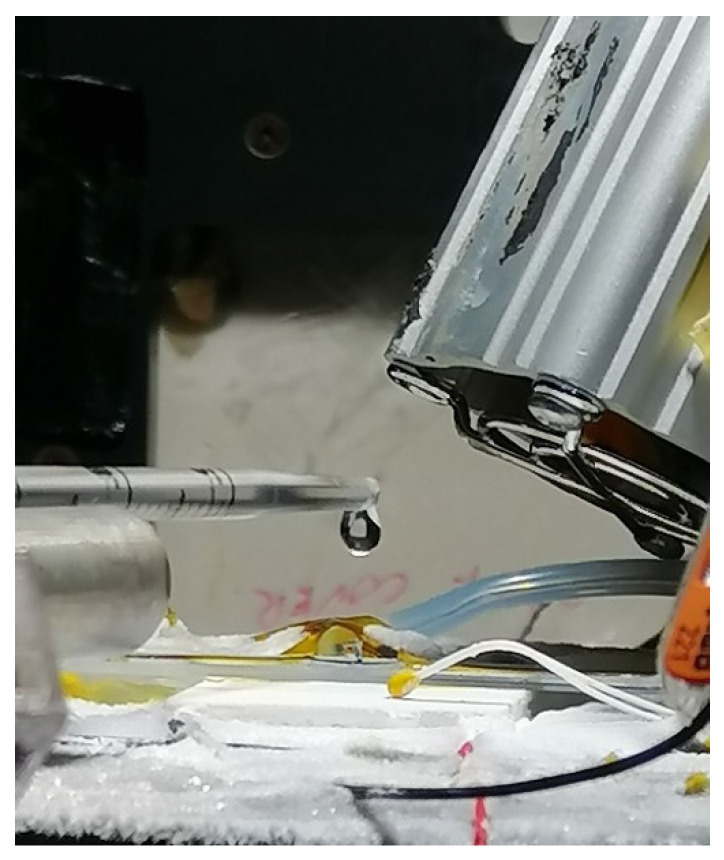
Example of drop injection (side view).

**Figure 4 micromachines-15-00504-f004:**
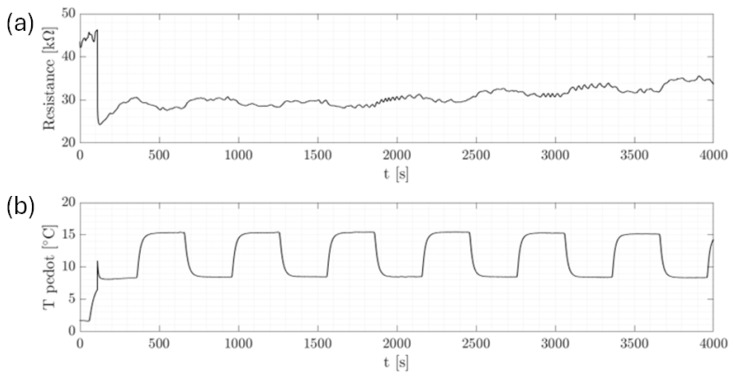
Temperature-controlled chamber experiment. Single drop. T${\;\;}_{cold}$ = 8 ${\;\;}^{\circ }$C, T${\;\;}_{chamber}$ = 15 ${\;\;}^{\circ }$C. (**a**) Resistance behavior. (**b**) Temperature behavior of the sensing element.

**Figure 5 micromachines-15-00504-f005:**
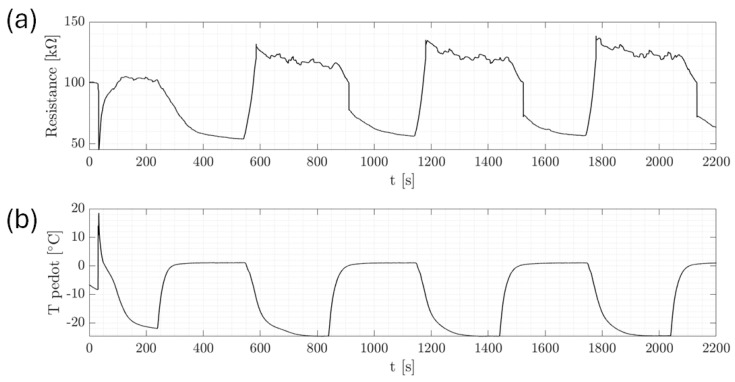
Characteristic water and ice signals at T${\;\;}_{cold}$ = −25 ${\;\;}^{\circ }$C and T${\;\;}_{chamber}$ = 15 ${\;\;}^{\circ }$C. (**a**) Resistance behavior. (**b**) Temperature behavior of the sensing element.

**Figure 6 micromachines-15-00504-f006:**
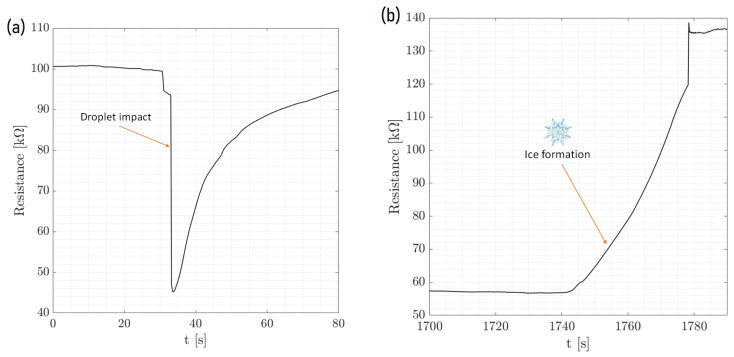
Temperature-controlled chamber experiment. (**a**) Single drop (zoom). T${\;\;}_{cold}$ = −25 ${\;\;}^{\circ }$C, T${\;\;}_{chamber}$ = 15 ${\;\;}^{\circ }$C. (**b**) Ice formation.

**Figure 7 micromachines-15-00504-f007:**
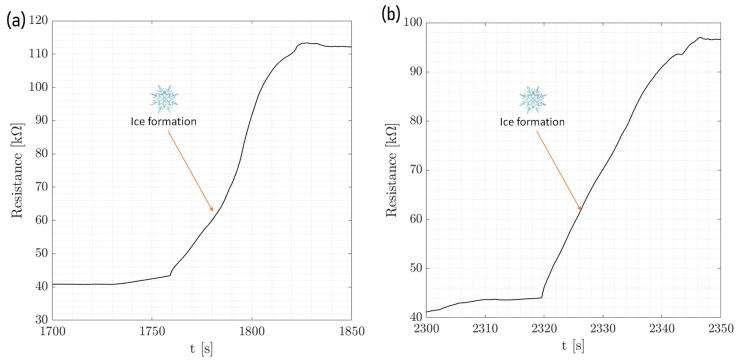
Temperature-controlled chamber experiment. Single drop (zoom). (**a**) T${\;\;}_{cold}$ = −20 ${\;\;}^{\circ }$C, T${\;\;}_{chamber}$ = 15 ${\;\;}^{\circ }$C. (**b**) $T_{cold}$ = −15 ${\;\;}^{\circ }$C, T${\;\;}_{chamber}$ = 15 ${\;\;}^{\circ }$C.

**Figure 8 micromachines-15-00504-f008:**
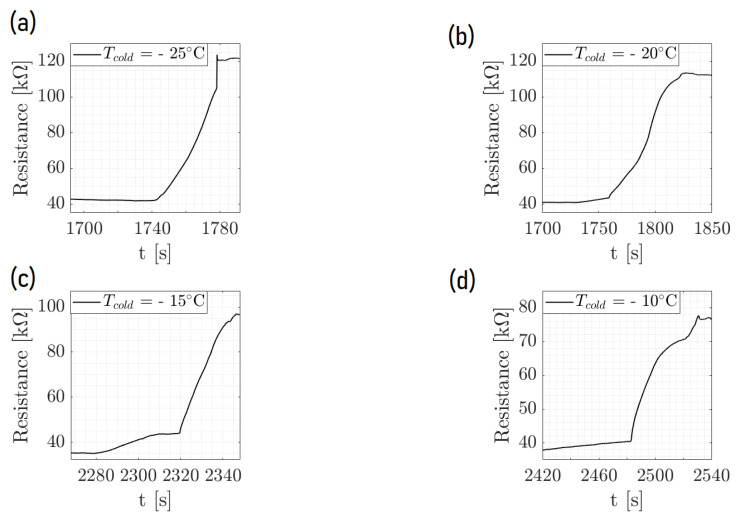
Temperature-controlled chamber experiment with T${\;\;}_{chamber}$ = 15 ${\;\;}^{\circ }$C illustrating the resistance change due to ice formation at a fixed chamber temperature of 15 °C. The subfigures (**a**–**d**) demonstrate the sensor’s response over time at various cold junction temperatures: (**a**) −25 °C, (**b**) −20 °C, (**c**) −15 °C, and (**d**) −10 °C. The sharp increase in resistance indicates the onset of ice formation on the sensor surface.

**Figure 9 micromachines-15-00504-f009:**
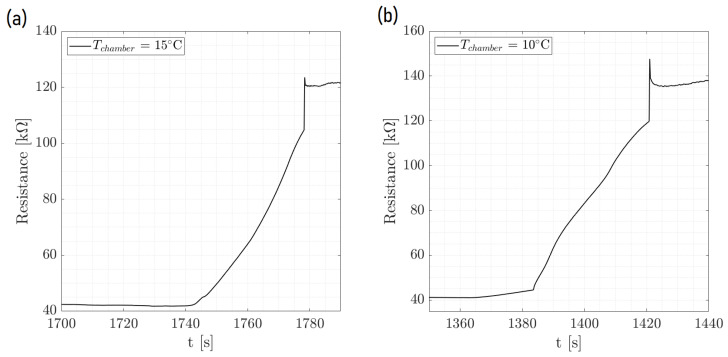
Response of the ice detection sensor at (**a**) ($T_{chamber}$ = 15 °C and (**b**) $T_{chamber}$ = 10 °C with $T_{cold}$ = −20 °C.

**Figure 10 micromachines-15-00504-f010:**
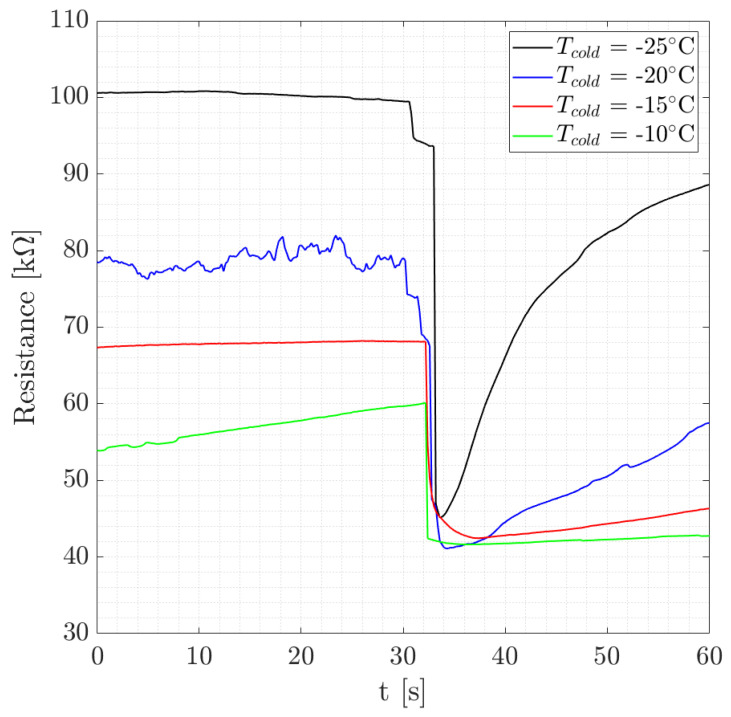
Sensor resistance behavior at different $T_{cold}$ levels with $T_{chamber}$ = 15 ${\;\;}^{\circ }$C.

**Table 1 micromachines-15-00504-t001:** Climatic chamber test matrix.

T Cold (${\;\;}^{\circ }$C)	Number of Iterations
−25 to −20	166
−20 to −15	45
−15 to −10	84
−10 to −5	80
−5 to 0	51

**Table 2 micromachines-15-00504-t002:** Comparison of the ice signals’ amplitudes.

$T_{\mathrm{cold}}$ (${\;\;}^{\circ }$C)	$\Delta R_{\mathrm{ICE}}$ (kΩ)
−25	82
−20	70
−15	52
−10	37

**Table 3 micromachines-15-00504-t003:** Comparison of the water signals’ amplitudes.

$T_{\mathrm{cold}}$ (${\;\;}^{\circ }$C)	$|\Delta R_{\mathrm{WATER}}|$ (kΩ)
−25	54
−20	38
−15	23
−10	18

## Data Availability

All system protocols are publicly accessible and can be found at the Université Libre de Bruxelles.

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
