# Peer review of "Environmental Chamber Characterization of an Ice Detection Sensor for Aviation Using Graphene and PEDOT:PSS"

_micromachines, 2024, doi:10.3390/mi15040504_

Round 1

Reviewer 1 Report

Comments and Suggestions for Authors

In this manuscript, a graphene-based ice detecting system was developed and tested in an environmental chamber. A variety of climatic conditions including the temperature and humidity levels were considered for the flat graphene-based sensor, and the results showed a good performance in ice detection. The manuscript is worthy of droplet icing measurement. However, it seems that the authors did not understand the phenomenon of aircraft icing and the need for the detection of the aircraft ice.  A major revision is required before the next step.

1. Aircraft icing is caused by the impact of supercooled water droplets. The temperature around the aircraft is always below 0 degree.

2. The introduction is very poorly prepared. The description of aircraft icing and the details of the traditional ice detection methods should be presented.

3. The principle of ice detection and analysis of detection errors need to be provided.

4. The scope of application of the detector and the constraints on its use need to be given.

Author Response

We sincerely thank Reviewer 1 for the constructive feedback, which has greatly contributed to the refinement of our manuscript. In response to the comments, we have thoroughly addressed each point, providing detailed explanations and making the necessary revisions to enhance our study's clarity and depth. Please find attached our point-by-point responses and the revised manuscript for your consideration. We hope our efforts adequately address the concerns raised and contribute to the improvement of our work.

Reviewer 2 Report

Comments and Suggestions for Authors

Dear authors,

This is a relevant work that faces a relevant topic, aeronautic icing sensing. The study is well defined, the methods are suitable and properly described, but the study objectives should be highlighted and more clearly diclosed in the final part of introduction. Discussions and conclussions are well supported.

Just a few comments:

·         In the Introduction section I miss more references and background of previous work, as it was done in the authors previous work of reference number 9

·         It is recommended write the full name of acronyms in first time in the text (i.e. PEDOT, PPS, PVDF, etc.)

·         Perhaps it is common in this journal, but the info of 2.1 and 2.2 used to be included in Introduction section as a part of needed background to understand the specific topic.

·         Please, check the Figure references along the text, there are some errors (i.e. line 117, 160, etc.)

·         Line 127: Please, specify where or which are “your facilities”.

·         Line 148: in the study 24 µl droplets were used. It will be useful also indicate the volumen diameter, since this is the reference parameter used in atmospheric icing literatura.

·         Just a comment, it is usually to publish the work in increasing complexity order. Why have the authors publish the wind tunnel results in first place, and then the lab testing?

Author Response

We deeply appreciate Reviewer 2's insightful and constructive feedback, which has been crucial in refining and enhancing our manuscript. The comments received were thoroughly reviewed and addressed, leading to significant improvements in our work's clarity, depth, and overall quality. Attached, you will find a detailed response to each of the comments, showcasing the adjustments and additions made to the manuscript based on your valuable suggestions. Your expertise and guidance have been instrumental in this revision process, and we eagerly await any further feedback you might have to offer.

Reviewer 3 Report

Comments and Suggestions for Authors

General comments;

The state-of-the-art should be more elaborated, the authors mostly refer to their own work (references 7-9).

The presentation of the figures lacks consistency. Some figures are labeled (a) and b)..) but most figures are unlabeled. And most figure captions lack sufficient information about what the figures describe.

In conclusions and discussion, the results need to be discussed in relation to existing ice detection sensors on the market. Are the developed sensors more sensitive? More durable? What about the cost in relation to other alternatives?

Specific comments:

Page 2, line 63, define GOPS

Page 7, line 232, Tcold, cold should be in subscript

Page 8, figure 5 and 7, mark a) and b) in the figures

Page 9, figure 8, the figures need to be marked a) to d) and also described in the figure caption

Page 10, figure 9, the figures need to be marked a) to d) and also described in the figure caption

Page 10, figure 10, the figure caption is not inline with the content in the figure and with what is described in the text.

Ref 9, info missing

Author Response

We extend heartfelt thanks to Reviewer 3 for the comprehensive feedback and valuable suggestions, which have been instrumental in enhancing our manuscript. Each point has been diligently addressed to refine and clarify our contributions to the field. Please find attached our detailed point-by-point responses, reflecting the thorough consideration given to your insights. We are grateful for the opportunity to improve our work with your guidance and look forward to any further feedback.

Round 2

Reviewer 1 Report

Comments and Suggestions for Authors

The response in the revised manuscript is not convincing, and I do not recommend journal publication.

Round 3

Reviewer 1 Report

Comments and Suggestions for Authors

No comments left. It can be pulished in Micromachines.